# Valence State and Catalytic Activity of Ni-Fe Oxide Embedded in Carbon Nanotube Catalysts

**DOI:** 10.3390/nano14242004

**Published:** 2024-12-13

**Authors:** Jaekwang Lee, Heesoo Lee

**Affiliations:** School of Materials Science and Engineering, Pusan National University, Busan 46241, Republic of Korea; jaekwanglee@pusan.ac.kr

**Keywords:** CNT synthesis, charge transfer, active oxygen species, redox property, catalytic activity

## Abstract

The catalytic activity of Ni-Fe oxide embedded in CNTs was investigated in terms of valence states and active oxygen species. Ni-Fe oxides were prepared by the sol-gel combustion process, and Ni-Fe oxides embedded in CNT catalysts were synthesized by the catalytic chemical vapor deposition (CCVD) method. The lattice structure of the Ni-Fe oxide catalysts was analyzed, and the lattice distortion was increased with the addition of Fe. The specific surface areas and pore structures of the Ni-Fe oxides embedded in CNTs were determined through the BET method. The nano-sized Ni-Fe oxides embedded in CNTs were observed using morphology analysis. The crystallinity and defects of CNTs were analyzed by Raman spectroscopy, and the I_D_/I_G_ ratio of Ni_1.25_Fe_0.75_O/CNT was the lowest at 0.36, representing the high graphitization and low structural defects of the CNT surface. The valence states of Fe and Ni were changed by the interaction between catalysts and CNTs. The redox property of the catalysts was evaluated by H_2_-TPR analysis, and the H_2_ consumption of Ni_1.25_Fe_0.75_O/CNT was the highest at 2.764 mmol/g. The catalytic activity of Ni-Fe oxide embedded in CNT exhibited much higher activity than Ni-Fe oxide for the selective catalytic reduction of NOx with NH3 in the temperature range of 100 °C to 450 °C.

## 1. Introduction

Carbon nanotubes (CNTs) have attracted increased attention in various fields (e.g., catalysts, sensors, and electrodes) owing to their unique electronic, mechanical, and structural properties [1,2,3]. CNTs are mainly used as supporters in the field of catalysts, and CNTs have excellent electron and charge transfer as the π electron density is increased by the modified sp^2^ hybridization on the CNT surface [4,5]. The high charge transfer on the CNT surface causes a change in the valence state of the catalyst particles supported on the CNTs, which improves the active sites on the surface of the catalyst particles [6].

For metal oxide catalysts, the concentration of the oxygen vacancies and active oxygen species depends on the support materials [7,8]. The concentrations of the oxygen vacancies and active oxygen species change owing to charge transfer occurring at the interface between the support and catalyst particles [9]. CNTs, TiO_2_, Al_2_O_3_, etc., are mainly adopted as candidates for catalyst support, and CNTs exhibit improved charge transfer by high electron density through sp^2^ hybridization [10,11]. Kazakov, M.O. et al. prepared CoMo/CNT and CoMo/Al_2_O_3_ catalysts and studied the improved properties of the CoMo/CNT catalyst in terms of charge-transfer-assisted change in the valence state [12].

The acid oxidation method is widely utilized to use CNTs as supporters, and functional groups are formed on the surface of CNTs to support catalyst particles such as metals and metal oxides [13,14]. Strong oxidizing acids such as nitric acid (HNO_3_), sulfuric acid (H_2_SO_4_), and hydrochloric acid (HCl) are used for the functionalization of CNTs, but they can cause serious structural defects on the surface of CNTs [15,16]. Some researchers have studied a method of synthesizing CNTs in which transition metal particles, such as Fe, Ni, and Co, are encapsulated or embedded in CNTs without the acid oxidation method [17,18]. Wang et al. designed a method for synthesizing Fe nanoparticles embedded in CNTs on ZSM-5 support while growing CNTs using Fe catalysts [19].

Iron (Fe) is one of the most widely employed catalysts for CNT synthesis because of its abundance, non-toxicity, high thermal stability, and carbon solubility [20,21]. Fe-based catalysts can promote CNT growth, but they can also cause aggregation of catalyst particles and reduce the quality of CNTs [22,23]. The addition of nickel (Ni) affects the activity and stability of Fe-based catalysts and the structural characteristics of CNTs, such as defects and crystallinity [24,25]. The low structural defects of CNTs improve the electronic and charge transfer at the interface between the CNT surface and catalyst particles, and these influence the formation of active sites [26,27]. Therefore, it is important to study the change in the valence state and active site of catalyst particles due to the improved charge transfer between catalyst particles and CNTs grown by Ni-Fe oxide catalysts.

We investigated the change in the valence state and active sites of the catalysts by charge transfer between CNTs and Ni-Fe oxide. Nano-sized Ni-Fe oxides embedded in CNTs were synthesized, and the morphology and interplanar distance of the catalyst particles were observed by TEM analysis. The specific surface areas and the pore structures of the catalysts were determined using the BET method. The structural properties of the CNTs were analyzed using Raman spectroscopy, and XPS analysis was used to identify the changes in the valence states of Ni. The redox property of the catalysts was evaluated by changing the active sites using H_2_-TPR. De-NOx efficiency was examined by conducting a fixed-bed reactor test with NOx gas to confirm the enhanced charge transfer between Ni-Fe oxide catalysts and CNTs.

## 2. Materials and Methods

### 2.1. Catalyst Preparation

Ni-Fe oxides were synthesized by the sol-gel combustion process to grow CNTs, and Nickel nitrate hexahydrate (Ni(NO_3_)_2_∙6H_2_O, 99.999%, Merck, Darmstadt, Germany) and iron nitrate nonahydrate (Fe(NO_3_)_3_∙9H_2_O, 99.999%, Merck) were used as the metal oxide catalyst precursors. The precursors were weighed to achieve a Ni/Fe weight ratio of 0.75:1.25, 1.0:1.0, 1.25:0.75, and 1.50:0.50, respectively. The metal precursor was dissolved in distilled water with stirring for 12 h to prepare the homogeneous metal nitrate solutions. The solutions were treated in an ambient atmosphere at 500 °C for 4 h. Ni-Fe oxide embedded in CNT was synthesized by catalytic chemical vapor deposition with Ni-Fe oxide powders at 850 °C for 30 min, and carbon monoxide (CO) was employed as a carbon precursor. The samples were named Ni_0.75_Fe_1.25_O/CNT, Ni_1.0_Fe_1.0_O/CNT, Ni_1.25_Fe_0.75_O/CNT, and Ni_1.50_Fe_0.50_O/CNT, with the weight ratio of Ni/Fe, respectively, and Ni_1.25_Fe_0.75_O was used for comparison.

### 2.2. Characterization

X-ray diffraction patterns were obtained at 30 kV and 40 mA using an X-ray diffractometer (XRD, Cu, Kα, X’pert Pro MPD), and the microstructures of the catalysts were observed using field-emission transmission electron microscopy (FE-TEM, JEOL JEM-2100) (Nanoscience Instruments, Phoenix, AZ, USA). The pore structure of Ni-Fe oxide/CNT and specific surface area change according to CNT growth were calculated using the Brunauer–Emmett–Teller (BET) method. The pore size distribution was determined in the desorption branch by the Barrett–Joyner–Halenda (BJH) method. Raman spectra were obtained to analyze the structural characteristics of the CNTs using a Raman spectrometer (LabRam HR800, Horiba Jobin-Yvon, Palaiseau, France) with charged-coupled device (CCD) detection, and a 532 nm wavelength laser was used to scan the Raman spectra with a resolution of 0.28 cm^−1^. The chemical states of the catalysts were analyzed by X-ray photoelectron spectroscopy (XPS, ESCALAB250, VG scientific, Waltham, MA, USA). The XPS results were corrected with C 1s (BE = 284.6 eV), and curve fitting was conducted using a Gaussian peak shape after standard Shirley background correction. H_2_–temperature programmed reduction (H_2_-TPR) was conducted at 10 °C/min up to 800 °C in a 30 mL/min flow of 10 vol.% H_2_ in -N_2_, and CuO was used as a calibration reference to quantify the total amount of consumed H_2_. The catalytic activity was measured in a fixed-bed flow reactor into which 120 mg of material was inserted. The concentrations of the simulated gases were 1000 ppm NO, 1000 ppm NH_3_, and 5 vol% O_2_ and were balanced by N_2_, with a total flow rate of 100 mL/min, which corresponds to a GHSV of 50,000 h^−1^. The feed gases were mixed and preheated in a chamber before entering the reactor. Catalytic activity was recorded after stabilizing for 80 min.

## 3. Results and Discussion

Figure 1 shows the XRD patterns of the catalysts; the spectra of all catalysts showed a peak at 2θ = 26.2°, which corresponds to the lattice plane (0 0 2) and is ascribed to graphite 2H (ICSD No. 98-005-3781). For all the catalysts in Figure 1, iron oxide (ICSD No. 96-100-8726) peaks were identified along with nickel oxide (ICSD No. 98-005-3781) peaks. This implies that CNTs were formed by graphitization induced by Fe-Ni oxide catalysts [18,28]. The intensity of the graphite (0 0 2) plane for Ni_1.25_Fe_0.75_O/CNT was confirmed to be the highest, representing a high yield of CNT by Ni-Fe oxide catalysts [29]. As shown in Figure 1b, the peak corresponding to the NiO (2 2 2) plane was shifted according to the content of iron, with the lowest angle shifted at Ni_1.25_Fe_0.75_O/CNT, indicating that the interplanar distance was narrowed. The peak shift implies the lattice expansion of NiO due to atomic radius difference, formation of oxygen species, and oxygen vacancies due to the change in the valence state of metal ions [30]. Since the atomic radii of Fe^3+^ and Ni^2+^ are the same at 0.069 nm, the lattice expansion of the NiO (2 2 2) plane can be caused by the repulsive force due to the formation of oxygen species or oxygen vacancies [31,32]. The average grain sizes of NiO were estimated to be 24 nm, 19 nm, 12 nm, and 21 nm for Ni_0.75_Fe_1.25_O/CNT, Ni_1.0_Fe_1.0_O/CNT, Ni_1.25_Fe_0.75_O/CNT, and Ni_1.50_Fe_0.50_O/CNT, respectively, using the Scherrer formula to consider the full width at half maximum (FWHM) of the peaks of NiO.

The surface area and pore size distribution of the catalysts were analyzed using the N_2_ adsorption–desorption measurements. All catalysts exhibited reversible type VI isotherms and H3 hysteresis loops at relative pressures of 0.5–1.0, as shown in Figure 2a; these indicate the mesoporous structure of the catalysts [33]. The specific surface areas of Ni_0.75_Fe_1.25_O/CNT, Ni_1.0_Fe_1.0_O/CNT, Ni_1.25_Fe_0.75_O/CNT, and Ni_1.50_Fe_0.50_O/CNT were calculated as 62.80 m^2^/g, 129.19 m^2^/g, 140.61 m^2^/g, and 80.56 m^2^/g, respectively. The specific surface area of Ni_1.25_Fe_0.75_O/CNT was the highest, which might be due to the promoted growth of CNTs by nano-sized Ni-Fe oxide particles [34]. The specific surface area increased as the aggregation of oxide catalyst particles was relieved by increasing the Fe content [20]. The specific surface area of Ni_0.75_Fe_1.25_O/CNT was reduced. The increased Fe content can easily cause the agglomeration of catalyst particles due to its magnetic properties [35,36], and the agglomerated catalyst particles make it difficult to grow CNTs, and the pores become clogged, resulting in a decrease in specific surface area [35,37]. The mesoporous structure was confirmed by the BJH desorption pore size distribution in Figure 2b, which shows that the Fe-Ni nanoparticles embedded in CNTs contain numerous mesopores with pore sizes in the range of 5–12 nm. The cumulative desorption pore volumes were 0.12 cm^3^/g, 0.15 cm^3^/g, 0.23 cm^3^/g, and 0.14 cm^3^/g for Ni_0.75_Fe_1.25_O/CNT, Ni_1.0_Fe_1.0_O/CNT, Ni_1.25_Fe_0.75_O/CNT, and Ni_1.50_Fe_0.50_O/CNT, respectively. The large pore volume and mesopore structure of Ni_1.25_Fe_0.75_O/CNT facilitate the improved electron and charge transfer, exposing more active sites on the surface of reactive species, which in turn enhances its electrochemical activity [38,39].

TEM analysis was conducted to identify the morphology and interplanar distance of the catalysts; Figure 3 displays the TEM images and energy dispersive spectroscopy (EDS) mapping of Ni-Fe oxide/CNT. Ni-Fe oxide nanoparticles of about 30 nm, 27 nm, 15 nm, and 25 nm in Ni_0.75_Fe_1.25_O/CNT, Ni_1.0_Fe_1.0_O/CNT, Ni_1.25_Fe_0.75_O/CNT, and Ni_1.50_Fe_0.50_O/CNT are observed in Figure 3. It was observed that MWCNTs with interlayer distances of 3.4 Å were grown on all catalysts, and it was confirmed that they were grown by the Ni-Fe oxide catalyst. The particle size increased as the Fe content increased, and the particles were agglomerated, indicating that the impregnation of excess Fe can lead to the agglomeration of the metal particles due to their magnetic properties [40].

The degree of crystallinity and defects on the CNT surface were investigated by Raman spectroscopy; Figure 4 shows the Fe-Ni oxide-embedded CNT spectrum in the range of 750–2250 cm^−1^. The Raman spectra of all catalysts displayed two characteristic bands at about 1332 cm^−1^ (D band) and 1580 cm^−1^ (G band). The D band is assigned to the out-of-plane vibrations of sp^3^-hybridized carbon atoms, which represent the disordered graphitic planes and defects in carbonaceous materials, while the G band is attributed to the in-plane vibrations of the sp^2^ carbon atoms [41,42]. The I_D_/I_G_ ratio is typically used to describe the degree of structural defects in carbon materials, and the defects in CNTs are directly related to catalytic activity because they increase the path of electron and charge transfer [21,43]. The I_D_/I_G_ ratios of Ni_0.75_Fe_1.25_O/CNT, Ni_1.0_Fe_1.0_O/CNT, Ni_1.25_Fe_0.75_O/CNT, and Ni_1.50_Fe_0.50_O/CNT were calculated to be 1.43, 0.97, 0.36, and 0.82, respectively. The low I_D_/I_G_ ratio of Ni_1.25_Fe_0.75_O/CNT signified a high degree of graphitization and low defects, which may enhance electron and charge transfer on the CNT surface.

Table 1 shows the surface atom concentration and XPS fitting results of the catalysts, and Figure 5 shows the Ni 2p, Fe 2p, and O 1s spectra for each catalyst and the binding energy according to the Ni ratio. The Ni 2p spectrum is divided into Ni 2p3/2 and Ni 2p1/2. The peaks at 852.9 eV and 869.3 eV were assigned to Ni^0^ 2p3/2 and Ni^0^ 2p1/2, and these correspond to the metallic state of nickel, and the peaks at 854.7 eV and 872 eV were attributed to Ni^2+^ 2p3/2 and Ni^2+^ 2p1/2 [44]. The two peaks at 710.8 eV and 724.0 eV were ascribed to the binding energy of Fe^3+^ 2p3/2 and Fe^3+^ 2p1/2, respectively, and the two peaks at 706.8 eV and 720.1 eV in the Fe 2p spectrum were attributed to the zero-valence state Fe [45].

The XPS spectra of O 1s in Figure 5 can be represented by the fitting curves of lattice oxygen (O_latt_) at 532.4 eV and oxygen vacancies (O_vac_) at 530.9 eV and 533.2 eV [46]. The lattice distortion identified in the XRD diffraction patterns of the Fe-Ni oxide/CNT catalysts is attributed to the intra-lattice repulsion caused by the formation of oxygen vacancies. The O_vac_/O_latt_ of Ni_1.25_Fe_0.75_O/CNT also showed the highest value, confirming that the most oxygen vacancies were formed on the catalyst surface. The change in oxygen vacancy formation on the NiO surface with Fe addition is judged to be caused by the difference in the valence states of Fe^3+^ and Ni^2+^, as Fe and Ni form the mixed oxide [47,48]. Oxygen vacancies can form or capture active oxygen species, which are chemical species including oxygen, such as O_2_^2−^ and O^−^ [49,50]. They serve as active sites during catalytic reactions because of their higher mobility compared to lattice oxygen.

H_2_-TPR analysis was performed to estimate the redox properties of the catalysts, and the resulting H_2_-TPR profiles of Ni-Fe oxide/CNT are shown in Figure 6. The catalysts showed different reduction processes based on the weight ratio of Ni. A weak reduction peak was identified at about 300 °C, and a broad peak was observed at about 440 °C for all catalysts, which corresponded to the reduction processes Fe_2_O_3_→Fe_3_O_4_→FeO and Ni^2+^→Ni^0^ [51]. In 0.75NiO/CNT and 1.0NiO/CNT, weak peaks were observed at about 550 °C, which corresponded to the reduction process FeO→Fe [51]. The reduction peaks of Ni-Fe oxide molecules embedded in CNTs were shifted to a lower temperature compared to Ni1.25Fe0.75O, and the peaks of Ni_1.25_Fe_0.75_O/CNT were also shifted to a lower temperature. The metal oxide supported on CNTs was easily reduced, and the oxygen in catalysts was more active, which reveals that the unique electronic properties of CNTs could promote charge transfer between metal atoms and CNTs, leading to enhancement in its redox potential [51]. The reduction peak was not observed at about 550 °C in Ni_1.25_Fe_0.75_O/CNT and Ni_1.50_Fe_0.50_O/CNT. Fe species are more difficult to reduce than the Ni species, and the reduction of the metal oxide catalyst was affected by the interaction between catalyst and supporter [52,53,54]. The interaction between catalyst and supporter weakened by decreasing the yield of CNTs as the content of Fe was increased. This made it difficult to reduce the catalyst. The total H_2_ consumption by Ni_0.75_Fe_1.25_O/CNT, Ni_1.0_Fe_1.0_O/CNT, Ni_1.25_Fe_0.75_O/CNT, Ni_1.50_Fe_0.50_O/CNT, and Ni_1.25_Fe_0.75_O were calculated to be 1.712 mmol/g, 2.374 mmol/g, 2.764 mmol/g, 1.856 mmol/g, and 1.349 mmol/g, respectively, as a result of integrating the H_2_ consumption. The highest H_2_ consumption by Ni_1.25_Fe_0.75_O/CNT was attributed to the enhanced redox property, which was due to the increased number of active sites on the catalyst surface due to the promoted charge transfer between the CNTs and Ni-Fe oxide nanoparticles [55].

The catalytic activity of Fe-Ni oxide embedded in CNT was examined using the SCR reaction for the removal of nitrogen oxides. The nitrogen oxide removal efficiency was measured using a fixed-bed reactor. The nitrogen oxide removal efficiency was measured from 100 to 450 °C (intervals of 50 °C) after being maintained at each specific temperature for 1 h 30 m to stabilize the efficiency at each condition. Figure 7 shows the NO_x_ conversion efficiency of Ni-Fe oxide embedded in CNT and Ni_1.25_Fe_0.75_O. Typical Fe oxide catalysts have low nitrogen oxide removal efficiency at temperatures below 350 °C as a single component [56,57]. The Ni-Fe oxide embedded in CNT exhibited enhanced efficiency over the entire temperature range compared to Ni-Fe oxide, which was reported to improve catalytic performance due to the increased active oxygen species generated by enhanced charge transfer between the oxide catalyst and CNT [58,59]. The nitrogen oxide removal efficiency of Ni_1.25_Fe_0.75_O/CNT improved by approximately 30% in the range of 200–250 °C compared with Ni_1.25_Fe_0.75_O. This indicates that the catalytic activity of Ni-Fe oxide embedded in CNT was increased by the formation of oxygen vacancies and active oxygen species since Ni^2+^ and Fe^3+^ were easily reduced by the enhanced charge transfer between Ni-Fe oxides and CNTs [55].

## 4. Conclusions

We investigated the catalytic activity of Ni-Fe oxides embedded in CNTs in terms of their valence state and active oxygen species. Ni-Fe oxides were prepared by the sol-gel combustion process, and Ni-Fe oxides embedded in CNT catalysts were synthesized by the CCVD method to improve the charge transfer between metal oxide and CNT without the acid oxidation method. Ni_1.25_Fe_0.75_O/CNT exhibited the highest graphite diffraction intensity and specific surface area, indicating the high yield of CNT by the Ni-Fe oxide catalyst. The nano-sized Ni-Fe oxide embedded in CNT was observed by morphology analysis, and the particle size of the oxides was increased due to the magnetic properties of Fe. The crystallinity and defects of the CNTs were analyzed using Raman spectroscopy, and the I_D_/I_G_ ratio of Ni_1.25_Fe_0.75_O/CNT was the lowest at 0.36, representing the high graphitization and low structural defects on the CNT surface. The valence states of the Ni-Fe oxides changed as the charge transfer between the oxides and CNTs was promoted by the low structural defects and high yield of CNTs. The enhanced redox properties of Ni_1.25_Fe_0.75_O/CNT were identified by the highest H_2_ consumption, which was calculated to be 2.764 mmol/g. The metal oxides were easily reduced by embedding CNTs, and the active oxygen species were increased, enhancing the redox potential of the catalysts. The nitrogen oxide removal efficiency of Ni_1.25_Fe_0.75_O/CNT was improved by approximately 30% in the range of 200–250 °C compared with Ni_1.25_Fe_0.75_O. The catalytic activity of Ni-Fe oxide/CNT was increased by the formation of active oxygen species since Ni^2+^ and Fe^3+^ were easily reduced by the improved charge transfer between oxides and CNTs.

## Figures and Tables

**Figure 1 nanomaterials-14-02004-f001:**
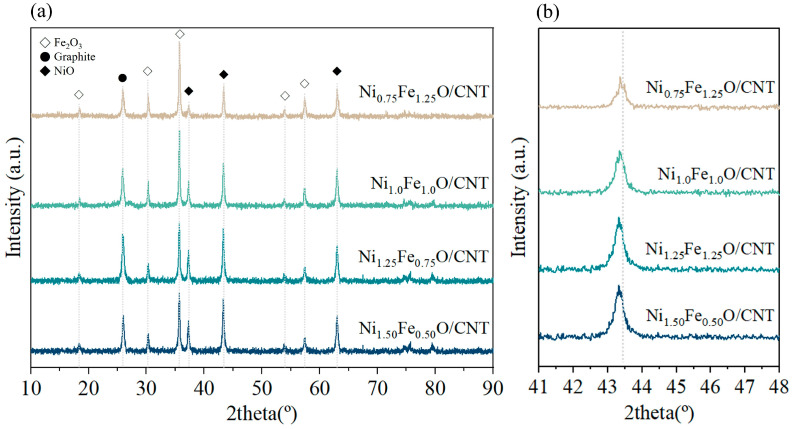
X-ray diffraction pattern analysis of Ni-Fe oxide/CNT (**a**) and with the magnifications of the NiO (2 2 2) plane region (**b**).

**Figure 2 nanomaterials-14-02004-f002:**
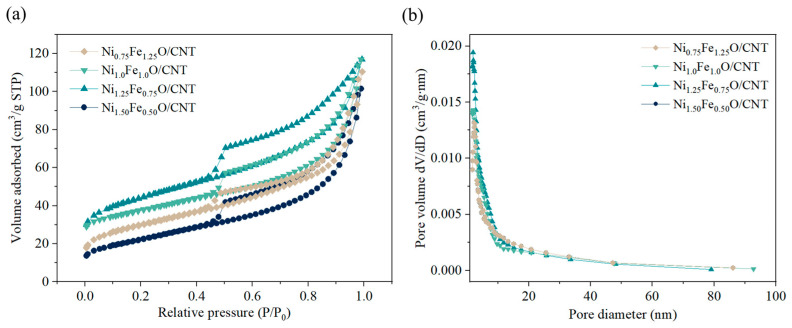
N_2_ adsorption–desorption isotherms (**a**) and pore distribution (**b**) of Ni-Fe oxide/CNT.

**Figure 3 nanomaterials-14-02004-f003:**
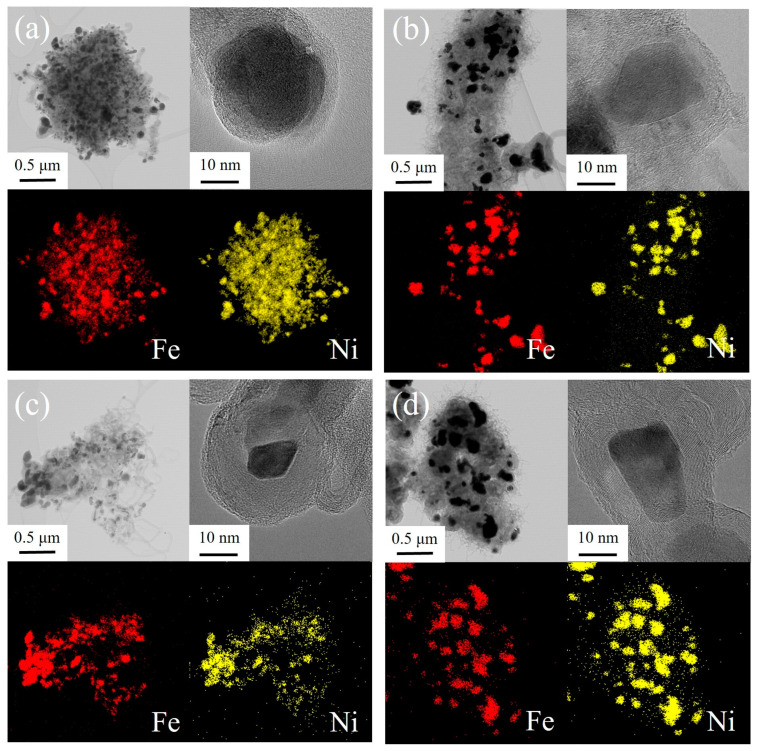
TEM images and EDS mapping of Ni_0.75_Fe_1.25_O/CNT (**a**), Ni_1.0_Fe_1.0_O/CNT (**b**), Ni_1.25_Fe_0.75_O/CNT (**c**), and Ni_1.50_Fe_0.50_O/CNT (**d**).

**Figure 4 nanomaterials-14-02004-f004:**
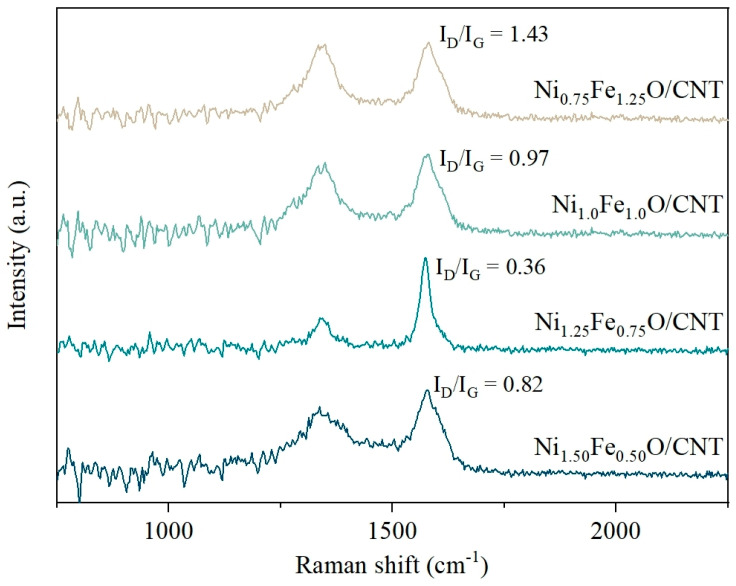
Raman spectra of Ni-Fe oxide/CNT.

**Figure 5 nanomaterials-14-02004-f005:**
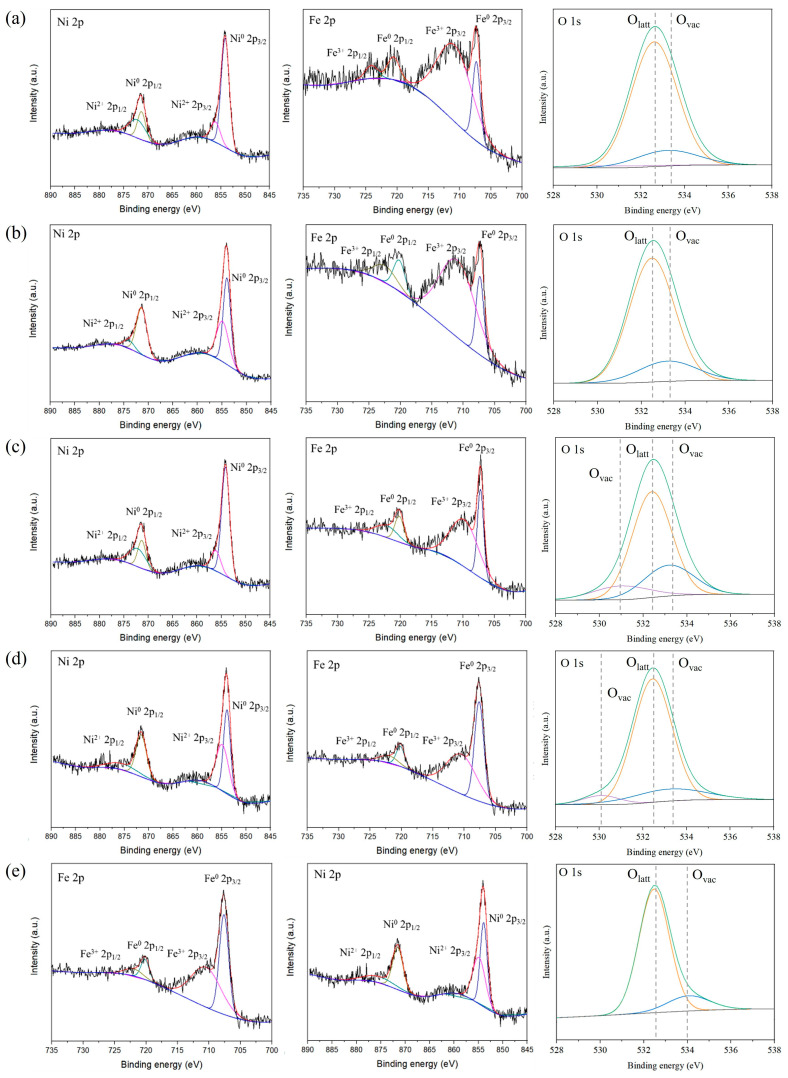
XPS spectra of Fe 2p, Ni 2p, and O 1s for Ni_0.75_Fe_1.25_O/CNT (**a**), Ni_1.0_Fe_1.0_O/CNT (**b**), Ni_1.25_Fe_0.75_O/CNT (**c**), Ni_1.50_Fe_0.50_O/CNT (**d**), and Ni_1.25_Fe_0.75_O (**e**).

**Figure 6 nanomaterials-14-02004-f006:**
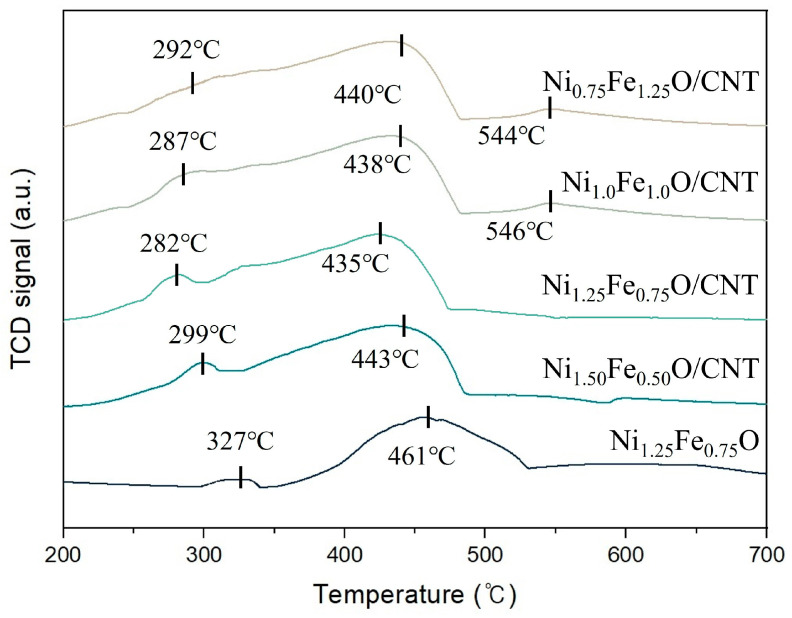
H_2_-TPR profiles of the Ni-Fe oxide/CNT.

**Figure 7 nanomaterials-14-02004-f007:**
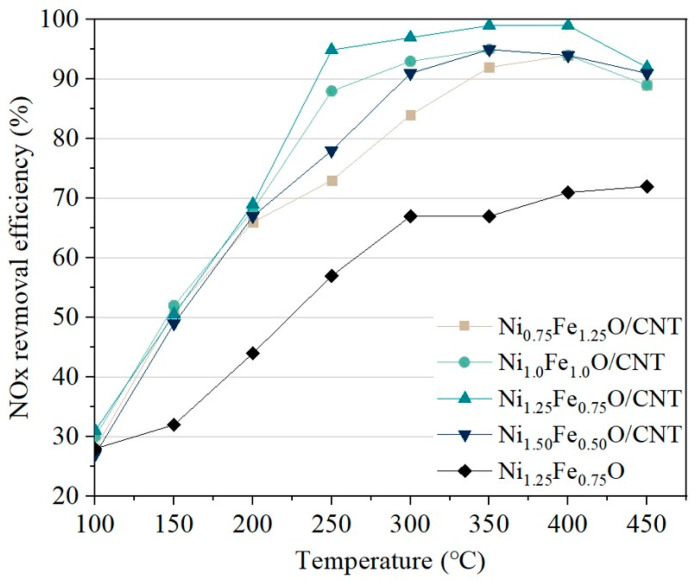
Comparison of NOx conversion efficiency of Ni-Fe oxide in embedded CNT and Ni-Fe oxide. Reaction conditions: [NO] = 1000 ppm or [NH_3_] = 1000 ppm, [O_2_] = 5 vol%, N_2_ balance and GHSV = 5000 h^−1^.

**Table 1 nanomaterials-14-02004-t001:** Surface atomic concentration and peak fitting results of the Ni-Fe oxide/CNT from XPS analysis.

Sample	Surface Atomic Concentration (at. %)				Peak Fitting Results (%)	
	C	Fe	Ni	O	Ni^2+^/ (Ni^2+^ + Ni^0^)	O_vac_/ (O_latt_ + O_vac_)
Ni_0.75_Fe_1.25_O/CNT	95.32	1.71	0.42	10.78	25.32	21.47
Ni_1.0_Fe_1.0_O/CNT	95.43	1.24	1.31	11.06	30.90	22.54
Ni_1.25_Fe_0.75_O/CNT	96.23	1.42	1.34	10.42	43.26	32.75
Ni_1.50_Fe_0.50_O/CNT	95.76	0.75	1.84	10.84	33.89	27.43
Ni_1.25_Fe_0.75_O	-	11	26.64	62.37	24.47	19.22

## Data Availability

Data are available upon reasonable request.

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
