# Peer review of "Valence State and Catalytic Activity of Ni-Fe Oxide Embedded in Carbon Nanotube Catalysts"

_nanomaterials, 2024, doi:10.3390/nano14242004_

Round 1
Reviewer 1 Report
Comments and Suggestions for Authors
The authors prepared NiFe oxide embedded in CNT catalysts for catalytic nitrogen oxide removal. However, in Reviewer’s opinion, the structural features of final catalyst were not clearly defined, demonstrated and discussed, and more results and discussion of this paper should be provided. Therefore, after carefully evaluate this work, the reviewer suggest that this paper can be accepted in this journal after addressing the following issues.
1. All the descriptions of “embedded CNT” should be changed to “embedded in CNT” including the Title of this manuscript.
2. All the abbreviations for the final product should be NixFeyO/CNT, not the 0.75NiO/CNT, 1.0NiO/CNT etc.
3. The high-resolution TEM image for Figure 3d was totally same to that of Figure 3c, the authors should clarify it precisely.
4. For the comparison purpose, the sample of Ni1.25Fe0.75O should be prepared and discussed the full paper.
5. Which kind of Ni-Fe oxide was used in Figure 7 should be clarified.
6. The Conclusion Section should be highly condensed.
7. Usage of the English language and scientific terminology should be improved.
Author Response
Comment 1: All the descriptions of “embedded CNT” should be changed to “embedded in CNT” including the Title of this manuscript.
Response 1: Thanks for your informative comment. We revised all the descriptions of “embedded CNTs” to “embedded in CNTs” and marked the corrections in blue.
Comment 2: All the abbreviations for the final product should be NixFeyO/CNT, not the 0.75NiO/CNT, 1.0NiO/CNT etc.
Response 2: We revised all abbreviations with the Ni:Fe ratio as follows: Ni0.75Fe1.25O/CNT, Ni1.0Fe1.0O/CNT, Ni1.25Fe0.75O/CNT, and Ni1.50Fe0.50O/CNT. And the revised abbreviations were marked in blue.
Comment 3: The high-resolution TEM image for Figure 3d was totally same to that of Figure 3c, the authors should clarify it precisely.
Response 3: Thank you for the detailed comment. We precisely modified the TEM images in Figure 3 (d).
Comment 4: For the comparison purpose, the sample of Ni1.25Fe0.75O should be prepared and discussed the full paper.
Response 4: Thanks for your important comment. We prepared Ni1.25Fe0.75O to compare the change in valence state of Ni-Fe oxide and catalytic activity with Ni-Fe oxide embedded in CNT as you recommended, and we discussed the change in oxygen vacancy, redox property, and catalytic activity by CNT supporter. We added and revised the manuscript, and the revised parts were marked in blue.
Comment 5: Which kind of Ni-Fe oxide was used in Figure 7 should be clarified.
Response 5: Ni-Fe oxide is Ni1.25Fe0.75O, and we changed the abbreviation in Figure 7 and manuscript as we responded to 4th comment.
Comment 6: The Conclusion Section should be highly condensed.
Response 6: Thanks for your valuable feedback. We condensed the conclusion section for better understanding as follows, and the revised part was marked in blue:
We investigated the catalytic activity of Ni-Fe oxides embedded in CNTs in terms of their valence state and active oxygen species. Ni-Fe oxides were prepared by the sol-gel combustion process, and Ni-Fe oxides embedded in CNT catalysts were synthesized by the CCVD method to improve the charge transfer between metal oxide and CNT without the acid oxidation method. Ni1.25Fe0.75O/CNT exhibited the highest graphite diffraction intensity and specific surface area, indicating the high yield of CNT by the Ni-Fe oxide catalyst. The nano-sized Ni-Fe oxide embedded in CNT was observed by morphology analysis, and the particle size of the oxides was increased due to the magnetic properties of Fe. The crystallinity and defects of the CNTs were analyzed using Raman spectroscopy, and the ID/IG ratio of Ni1.25Fe0.75O/CNT was the lowest at 0.36, representing the high graphitization and low structural defects on the CNT surface. The valence states of the Ni-Fe oxides were changed as the charge transfer between the oxides and CNTs was promoted by the low structural defects and high yield of CNTs. The enhanced redox properties of Ni1.25Fe0.75O/CNT were identified by the highest H2 consumption, which was calculated to be 2.764 mmol/g. The metal oxides were easily reduced by embedding CNTs, and the active oxygen species were increased, enhancing the redox potential of the catalysts. The nitrogen oxide removal efficiency of Ni1.25Fe0.75O/CNT was improved by approximately 30 % in the range of 200-250 ℃, compared with Ni1.25Fe0.75O. The catalytic activity of Ni-Fe oxide/CNT was increased by the formation of active oxygen species since Ni2+ and Fe3+ were easily reduced by the improved charge transfer between oxides and CNTs.
Comment 7: Usage of the English language and scientific terminology should be improved.
Response 7: Native speaker and we carefully reviewed and revised the manuscript to modify and improve the English language and scientific terminology.
Reviewer 2 Report
Comments and Suggestions for Authors
In the paper entitled "Valence state and catalytic activity of Ni-Fe oxide embedded CNT catalyst", the Authors presented the catalytic activity of Ni-Fe oxides embedded CNTs regarding their valence state. The studies included H2 consumption measurements and the reduction of NOx. Although the quality of the manuscript is good, some minor improvements have to be implemented before publication:
1. Please enlarge all the figures for better quality.
2. XRD studies - Figure 1b - Is the peak intensity growing linearly? Please confirm.
3. BET studies - please explain the difference between 1.25NiO/CNT and 1.50NiO/CNT (approx. 60 m2/g) in the catalysts' surface area and pore size distribution.
4. Figure 6 - H2-TPR analysis - in the case of 1.25NiO/CNT, the "peak" at 312 oC doesn't correspond with the actual peak at the plot. Please explain.
5. H2-TPR analysis - Why the peak attributed to a reduction process: FeO→Fe wasn't observed in the lower Fe concentrations (1.25NiO/CNT and 1.50NiO/CNT)?
Comments on the Quality of English LanguageLine 187: "... lattice expansion caused by lattice expansion..." should be rewritten for clarity.
Author Response
Comments 1: Please enlarge all the figures for better quality.
Response 1: Thanks for your comment. We improved the resolution of all the Figures for better quality.
Comment 2: XRD studies - Figure 1b - Is the peak intensity growing linearly? Please confirm.
Response 2: The peak intensity in Figure 1b grew linearly by decreasing Fe content, and in other words, the peak intensity was affected by the Ni:Fe ratio. Peak intensity in XRD spectra was generally affected by the composition, crystal structure, crystalline size, etc[1, 2, 3]. In the mixed metal oxide, the peak intensity can change with chemical composition if new crystal structure was not formed[1, 3].
[1] Li, B., Mei, T., Du, S., & Zhang, W. (2020). Synthesis of Ni–Fe and Ni–Fe/ZrO2 composite coating and evaluation of its structural and corrosion resistance. Materials Chemistry and Physics, 243, 122595.
[2] Pasuk, I., Neațu, F., Neațu, Ș., Florea, M., Istrate, C. M., Pintilie, I., & Pintilie, L. (2021). Structural details of BaTiO3 nano-powders deduced from the anisotropic XRD peak broadening. Nanomaterials, 11(5), 1121.
[3] Nguyen, V. H., Nguyen, T. D., Bach, L. G., Hoang, T., Bui, Q. T. P., Tran, L. D., ... & Do, S. T. (2018). Effective photocatalytic activity of mixed Ni/Fe-base metal-organic framework under a compact fluorescent daylight lamp. Catalysts, 8(11), 487.
Comment 3: BET studies - please explain the difference between 1.25NiO/CNT and 1.50NiO/CNT
(approx. 60 m2/g) in the catalysts' surface area and pore size distribution.
Response 3: The increased Fe content can easily cause the agglomeration of catalyst particles due to its magnetic properties[1], and the agglomerated catalyst particles are difficult to grow CNTs, and the particles can clog the pores, resulting in a decrease in specific surface area[2, 3].
[1] Han, Y. J., & Park, S. J. (2017). Influence of nickel nanoparticles on hydrogen storage behaviors of MWCNTs. Applied Surface Science, 415, 85-89.
[2] Dai, L., Chen, Y., Liu, R., Li, X., Ullah, N., & Li, Z. (2021). CO2 hydrogenation to C5+ hydrocarbons over K‐promoted Fe/CNT catalyst: Effect of potassium on structure–activity relationship. Applied Organometallic Chemistry, 35(7), e6253.
[3] Tavasoli, A., Trépanier, M., Abbaslou, R. M. M., Dalai, A. K., & Abatzoglou, N. (2009). Fischer–Tropsch synthesis on mono-and bimetallic Co and Fe catalysts supported on carbon nanotubes. Fuel Processing Technology, 90(12), 1486-1494.
So, we added the following statements in the manuscript to provide a more specific interpretation of the BET analysis, and the revised parts were marked in blue:
The specific surface area of Ni0.75Fe1.25O/CNT was reduced. The increased Fe content can easily cause the agglomeration of catalyst particles due to its magnetic properties[35, 36], and the agglomerated catalyst particles are difficult to grow CNTs, and the pores become clogged, resulting in a decrease in specific surface area[37, 38].
Comment 4: Figure 6 - H2-TPR analysis - in the case of 1.25NiO/CNT, the "peak" at 312 oC doesn't
correspond with the actual peak at the plot. Please explain.
Response 4: Thanks for your important comment. We revised Figure 6, which shows H2-TPR profiles of catalysts, and all the reduction peaks were corresponded with the actual peaks.
Comment 5: H2-TPR analysis - Why the peak attributed to a reduction process: FeO→Fe wasn't
observed in the lower Fe concentrations (1.25NiO/CNT and 1.50NiO/CNT)?
Response 5:
Thanks for your informative question. It is reported Fe species are more difficult to reduce than the Ni species, and the reduction of metal oxide catalyst was affected by the interaction between catalyst and supporter [1, 2, 3]. The interaction between catalyst and supporter weakened by decreasing the yield of CNTs as the content of Fe was increased. Consequently, this made it difficult to reduce the catalyst.
[1] Fang, H., Zheng, J., Luo, X., Du, J., Roldan, A., Leoni, S., & Yuan, Y. (2017). Product tunable behavior of carbon nanotubes-supported Ni–Fe catalysts for guaiacol hydrodeoxygenation. Applied Catalysis A: General, 529, 20-31.
[2] Fang, C., Zhang, D., Shi, L., Gao, R., Li, H., Ye, L., & Zhang, J. (2013). Highly dispersed CeO2 on carbon nanotubes for selective catalytic reduction of NO with NH 3. Catalysis Science & Technology, 3(3), 803-811.
[3] Bayat, N., Rezaei, M., & Meshkani, F. (2016). Methane decomposition over Ni–Fe/Al2O3 catalysts for production of COx-free hydrogen and carbon nanofiber. International journal of hydrogen energy, 41(3), 1574-1584.
So, we added the following statements in the manuscript for better understanding, and the revised parts were marked in blue:
The reduction peak wasn’t observed at about 550 ℃ in Ni1.25Fe0.75O/CNT and Ni1.50Fe0.50O/CNT. Fe species are more difficult to reduce than the Ni species, and the re-duction of metal oxide catalyst was affected by the interaction between catalyst and supporter [53, 54, 55]. The interaction between catalyst and supporter weakened by decreasing the yield of CNTs as the content of Fe was increased.
Comment 6: Line 187: "... lattice expansion caused by lattice expansion..." should be rewritten for clarity.
Response 6: Thank you for the detailed comment. We revised the sentence, and the revised sentence was marked in blue as follows:
The lattice distortion identified in the XRD diffraction patterns of the Fe-Ni oxide/CNT catalysts is attributed to the intra-lattice repulsion caused by the formation of oxygen vacancies.
Round 2
Reviewer 1 Report
Comments and Suggestions for Authors
The authors have addressed all my comments and I suggest its publication without further revision.